# On-State Current Degradation Owing to Displacement Defect by Terrestrial Cosmic Rays in Nanosheet FET

**DOI:** 10.3390/mi13081276

**Published:** 2022-08-08

**Authors:** Jonghyeon Ha, Gyeongyeop Lee, Hagyoul Bae, Kihyun Kim, Jin-Woo Han, Jungsik Kim

**Affiliations:** 1Department of Electrical Engineering, Gyeongsang National University, Jinju 52828, Gyeongnam, Korea; 2Division of Electronics Engineering, and Future Semiconductor Convergence Technology Research Center, Jeonbuk National University, Jeonju 54896, Jeonbuk, Korea; 3Center for Nanotechnology, NASA Ames Research Center, Moffett Field, Santa Clara, CA 94035, USA

**Keywords:** displacement defect, nanosheet field-effect-transistor, cosmic ray, terrestrial radiation, technology computer-aided design (TCAD), Sentaurus device-QTX, scattering model, temperature effect

## Abstract

Silicon displacement defects are caused by various effects. For instance, epitaxial crystalline silicon growth and ion implantation often result in defects induced by the fabrication process, whereas displacement damage is induced by terrestrial cosmic radiation. Clustered displacement damage reportedly reduces the on-state current (*I_on_*) in ordinary MOSFETs. In the case of an extremely scaled device such as a nanosheet field-effect transistor (NS-FET), the impact of displacement defect size was analyzed on the basis of the NS dimensions related to the device characteristics. In this study, we investigated the effect of displacement defects on NS-FETs using technology computer-aided design; the simulation model included quantum transport effects. The geometrical conditions, temperatures, trap concentrations, and scattering models were considered as the variables for on-state current reduction.

## 1. Introduction

Nanosheet (NS) field-effect transistors (FETs) with surrounded-gate architecture have been scaled down further than fin field-effect transistors (FinFETs) for sub-5 nm technology nodes [1,2,3]. The fabrication of perfect (defect-free) silicon crystals is a challenging task; defects may be introduced into the crystal anytime during the ingot manufacturing process, wafer fabrication, or even on the fly. Moreover, the development of imperfect epitaxial growth of silicon NS and lateral straggle of impurities during ion implantation may introduce point defects, lattice misfit, and dislocations [4,5,6]. Furthermore, terrestrial radiation owing to alpha decay and cosmic-ray neutrons from outer space continuously tamper with packaged semiconductor devices [7,8]. Historically, the issue of radiation-induced displacement damage in semiconductors has been of interest primarily in space and military applications [9]. The terrestrial neutron spectrum of New York at sea level ranges from a few MeV to a few GeV, and the approximate flux of the peak regime between 1 and 100 MeV is 20/cm^2^·h [10]. Because the displacement threshold energy of silicon is 50 KeV, neutron-induced displacement damage accumulates over time. The event rate of heavy ion scattering for silicon is calculated by multiplying a heavy-ion flux, *ϕ* (heavy-ion/cm^2^·h); the atomic number density of the region of interest, *n* (atoms/cm^3^); and the cross section of scattering process, *σ* (1 barn = 10^−24^ cm^2^). Using cosmic ray data at the altitude of 20,000 km (GPS orbit) [11], a heavy ion is monoenergetic at 1 MeV, and its flux at a certain location is 1.44 × 10^13^ heavy-ion/cm^2^·h; the silicon atom count is 8 × 10^21^/cm^3^ for the silicon cubic, and the cross section of a 1 MeV neutron is 2 barns. Thus, the event rate becomes 2.5 × 10^11^/cm^3^·h. Considering the large size of semiconductor devices relative to that of a few silicon voids, the effects of terrestrial radiation are negligible and have generally been ignored. However, as the scaling of such devices approaches the fundamental limit, it is unclear whether displacement defects cause notable changes in the electrical behavior. As the NS thickness and channel length are scaled down to that of a few atoms, a single silicon vacancy cluster is sufficient to cause perturbations in the carrier transport. Additionally, the size of the silicon vacancy cluster may approximate the cross-sectional area of the NS.

In this study, we investigated the impact of cube-like defects on NS-FET technology with the aid of three-dimensional (3D) technology computer-aided design (TCAD) simulation. In particular, the effects of defect location and temperature with different trap concentrations were investigated. In addition, the dependence of defects on distinct NS widths and thicknesses was studied. Lastly, the contribution of the on-state current to the degradation mechanism was investigated for various scattering sources.

## 2. Materials and Methods

A Synopsys Sentaurus TCAD simulator was used in this study [12]. The 3D device structure of the NS-FET is illustrated in Figure 1a. The *I_d_*–*V_g_* characteristics reported in [1] were employed for calibration of the simulation model, as shown in Figure 1b The physical (*L_g_*) and effective (*L_eff_*) gate lengths are 12 and 8 nm, respectively. The three NSs are stacked and share a common source and drain. The top, middle, and bottom NSs are denoted by NS1, NS2, and NS3, respectively. For the device model, thin-layer mobility and Auger recombination models, a Shockley–Read–Hall model with doping dependence, and a Hurkx band-to-band tunneling model were adopted to consider quantum confinement in the short channel [13,14,15]. A more sophisticated carrier transport model is essential for the aggressively scaled node [16,17,18]. To consider the subband Boltzmann transport equation, Schrödinger equation, and Poisson equation (3D) coupled with the drift-diffusion transport model, a Sentaurus Device QTX instrument (quantum transport model of carrier quasi-ballistic transport) was utilized [19]. The displacement defect was expressed as a cubic trap cluster, and the defect size was 2 nm, with the accepter-like energy level at *E_c_* = −0.4 eV. The concentration of the displacement defect used was 8 × 10^21^ cm^−3^. There were 512 ea silicon lattices in the cubic trap cluster of 4 nm size (4 nm × 4 nm × 4 nm). The on-state current (*I_on_*) was extracted at *V_d_ = V_g_* = 0.7 [V].

## 3. Results and Discussion

The *I_d_*–*V_g_* characteristics with NS width (*D**_sheet_*) of 20 nm and thickness (*T_sheet_*) of 5 nm with and without a trap for different NSs are illustrated in Figure 2a The center location of the trap cluster was fixed at the exact center of each NS, adjacent to the source side. In a similar study based on FinFET technology, this position resulted in extreme displacement defects [7]. The trap at the top NS exhibited greater *I_on_* degradation than that at the bottom NS because the top layer, which was the closest junction to the contact, acted as the main current path. The impact of different locations of the trap in the vicinity of the source of NS1 is plotted in Figure 2b. As the trap moves from the edge of the NS1 sidewall to the center, the *I_on_* degradation intensifies, owing to the main carrier transport through the center of the NS [2,20]. Thus, the device in the volume inversion would experience maximum degradation if the defect formed at the center rather than the edge of the channel. To determine the defect tolerance for various physical dimensions of the NS, *D_Sheet_* and *T_sheet_* were varied while the position of the defect was kept constant. In this simulation, the trap was located adjacent to the source and center of NS1. As shown in Figure 2c, the *I_on_* degradation rate ((no trap device’s *I_on_*—trap device’s *I_on_*)/no trap device’s *I_on_* × 100%) increases from 18.8% (*D_Sheet_* = 20 nm) to 22.9% (*D_Sheet_* = 12 nm). This result is attributed to the greater vulnerability of the nanowire-like channel with a narrow channel width compared to the NS-like channel with a wide channel width, which is associated with the rise in the occupancy of the defect over the size of the channel as *D_Sheet_* decreases. Similarly, the *I_on_* degradation increases moderately with the reduction of the channel thickness. Although a thin and narrow channel is favorable for gate controllability and short-channel effect immunity, the blockage of the carrier conduction path owing to the displacement defect could have catastrophic effects.

In order to understand the results of Figure 2c, the electron density profiles cut at the XY-plane with and without a displacement defect are depicted in Figure 3. Compared to the electron density profile of the no-trap device, the electron density of the device with a trap decreases due to the displacement defect. The thick NS device exhibits a rate of electron density reduction lower than that of the control device, which results from the increase in the NS volume. Although the thick NS structure is unfavorable for electrostatic gate controllability, the blockage of the channel is more dominant on the *I_on_* degradation owing to the displacement defect in the narrow channel structure; (20,6), (20,5), and (12,5) had off-state current degradations of 32.29%, 31.03%, and 30.90%, respectively. The off-state current had more variability than the on-state current. However, a large *I_off_* variability in *V_g_* = 0 [V] and *V_d_* = 0.7 [V] did not necessarily matter because the absolute value of the leakage current was very small.

The *I_d_*–*V_g_* characteristics at different temperatures for the trap located at the source side of the NS1 layer are plotted in Figure 4a. Generally, as the temperature decreases, the subthreshold slope steepens as the off-state current decreases and the drive current increases in the silicon channel. The effect of the defect on device performance is smaller at lower temperatures. This is because the captured electrons increase with higher temperature due to abundant carrier generation. Thus, the current degradation becomes more severe as the temperature increases from 230 K to 350 K.

The rate of *I_on_* degradation with various trap concentrations and device temperatures is plotted in Figure 5. As the trap concentration increases, the rate of *I_on_* degradation dramatically increases. The *I_on_* degradation at 300 K is larger than that at 350 K, increasing from 8 × 10^18^ cm^−3^ to 8 × 10^20^ cm^−3^, because the additional free carriers generated at high temperature diminish the *I_on_* degradation due to the displacement defect. However, the *I_on_* degradation at 350 K is larger than that at room temperature (300 K) when the trap concentration is very large at 8 × 10^21^ cm^−3^. This is due to the fact that displacement defects are abundant to capture the additional free carriers generated by high temperature. Therefore, the temperature dependency of *I_on_* degradation differs with displacement defect density.

To investigate the primary cause of mobility degradation, the simulation was conducted using several scattering models. The *I_on_* degradations with various scattering models are displayed in Figure 6. The contribution of coulomb scattering (CO) is the greatest for *I_on_* degradation compared to that of surface roughness scattering (SR) or phonon scattering (PH). The influence of CO scattering causes carrier mobility degradation owing to interactions between the trapped and free electrons [21]. Furthermore, as the device is aggressively scaled down to decrease the channel length and thickness of the silicon oxide layer, the free-electron carriers are further affected by CO scattering [22]. In a short channel, the current fluctuation by PH and SR scattering was low, but the trap spread to the silicon oxide surface, so the current fluctuation is slightly higher for surface roughness scattering than phonon scattering. Phonon scattering is primarily dominant at high temperatures, and the quantity of the silicon lattice in the channel is minimal; thus, the PH model has the lowest impact among the scattering models. In addition, due to summation of Matthiessen’s rule of carrier mobility, the current change of all scattering is not manifested, as the summation of the current changes of another single scattering has a slightly higher current variation ratio than coulomb scattering, which generates the most current variation. The above result means that carrier trapping by the trap is the more-dominant mechanism compared to scattering variation in *I_on_* degradation.

## 4. Conclusions

In this study, we explored *I_on_* degradation due to displacement defects from cosmic rays in NS-FETs using TCAD simulation. The largest degradation of on-state current is 18.8%, where the displacement defect is located in the NS1 layer, at the center of the NS and near the source. The nanowire structure with a small diameter and thickness is more susceptible to degradation due to displacement defects compared to the nanosheet structure with a wide diameter. For the temperature effect, the pattern of *I_on_* degradation at high and low temperatures differs with concentrations of displacement defects. Coulomb scattering was determined as the major cause of degradation, compared with phonon scattering and surface roughness scattering.

## Figures and Tables

**Figure 1 micromachines-13-01276-f001:**
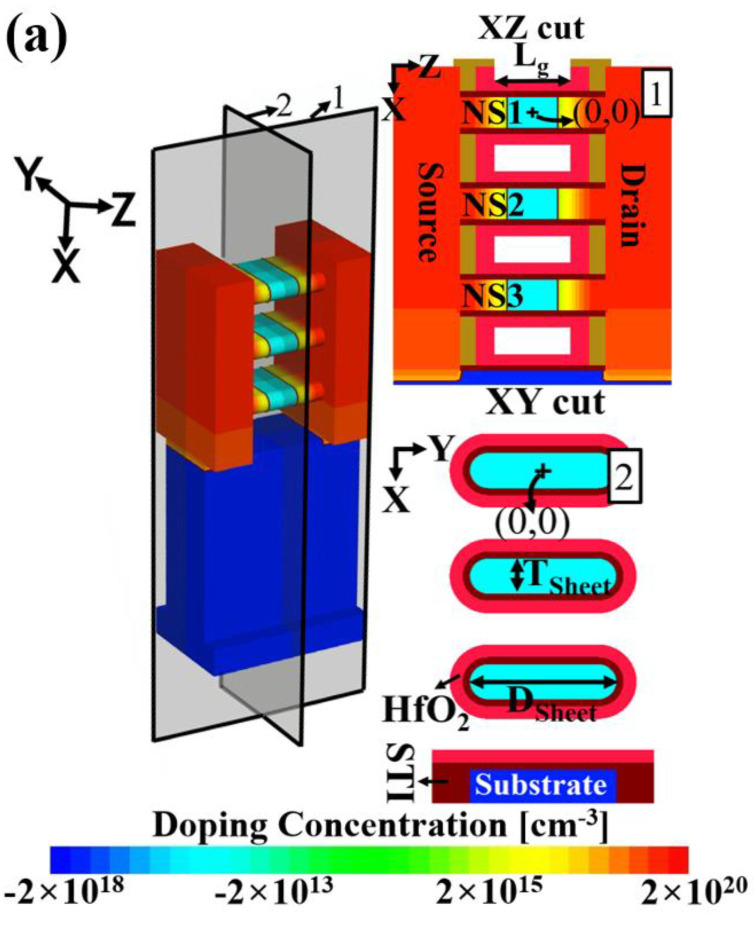
(**a**) 3D device structure with doping profile. Positive (+) and negative (−) signs in the concentration legend represent *n*− and *p*−type doping, respectively. (**b**) Drain current (*I**_d_*) versus gate voltage (*V_g_*) characteristics with calibration of experimental reference [1].

**Figure 2 micromachines-13-01276-f002:**
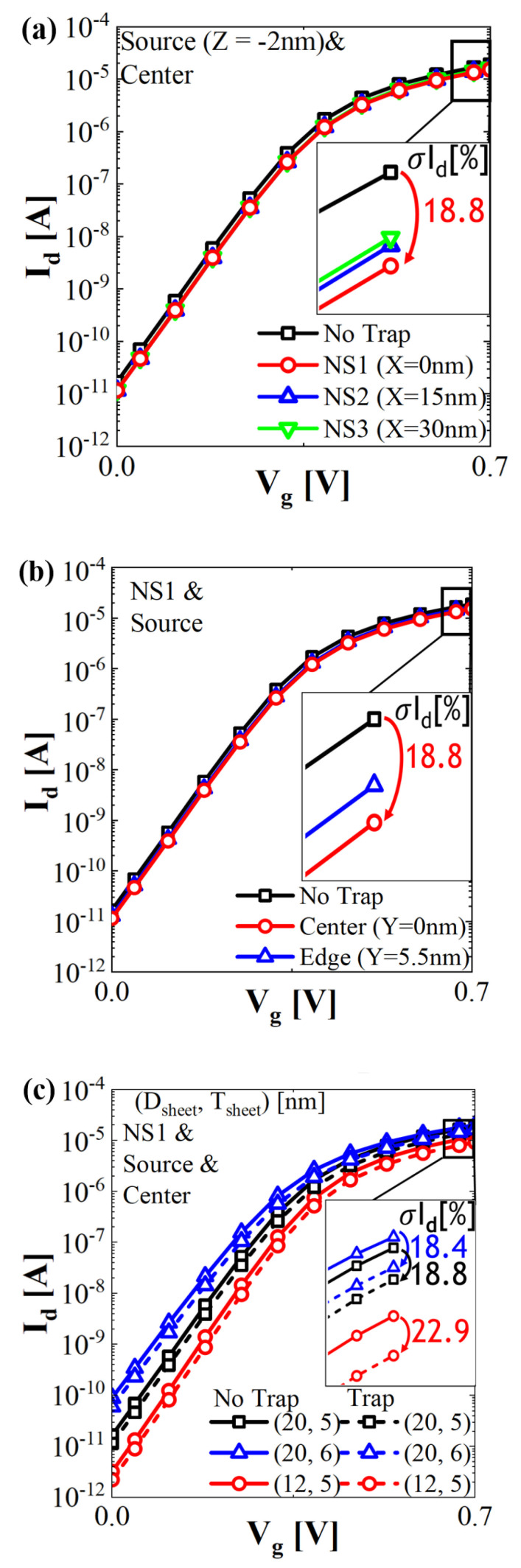
*I_d_*−*V_g_* characteristics without and with trap. (**a**) Different trap positions in various channel layers (X). (**b**) Different trap positions with regard to channel diameter direction (Y) in the NS1 layer. (**c**) Various diameters and thicknesses of channel. In the NS1 layer, trap location to channel length direction and sheet diameter direction are fixed at the source side and center, respectively.

**Figure 3 micromachines-13-01276-f003:**
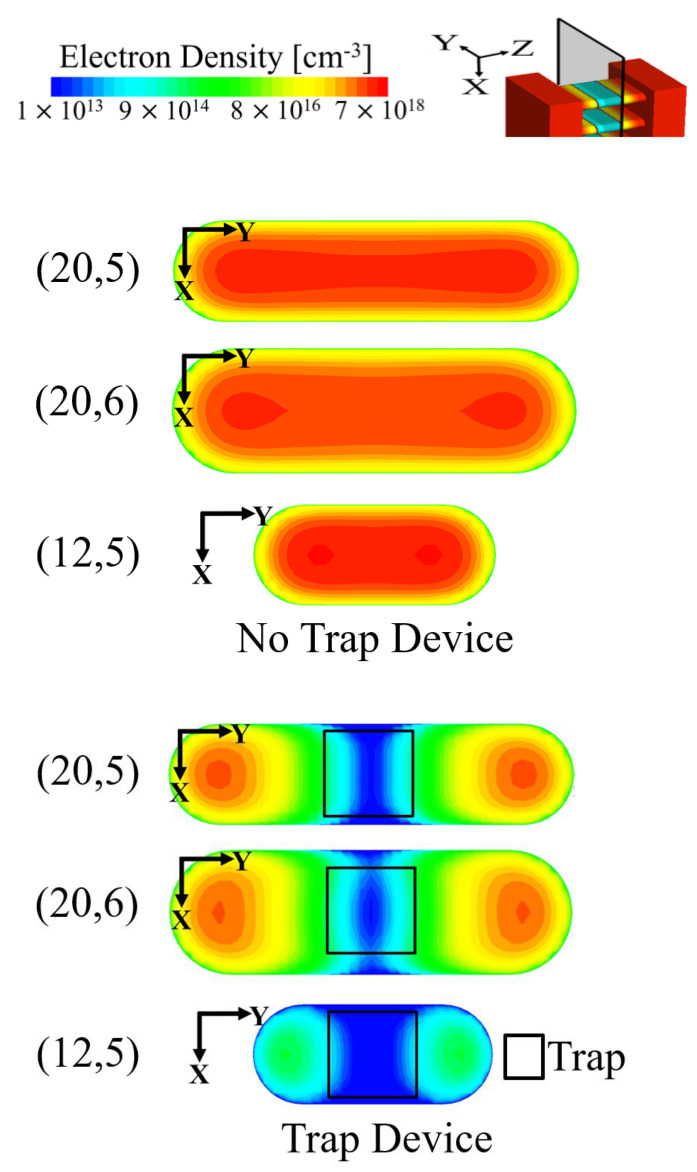
Electron density profiles of NS1 without and with displacement defect (source side, center of channel diameter in NS1 layer) along XY-plane at Z = −0.2 nm. (20,5), (20,6), and (12,5) denote the control, thick channel, and narrow channel devices, respectively.

**Figure 4 micromachines-13-01276-f004:**
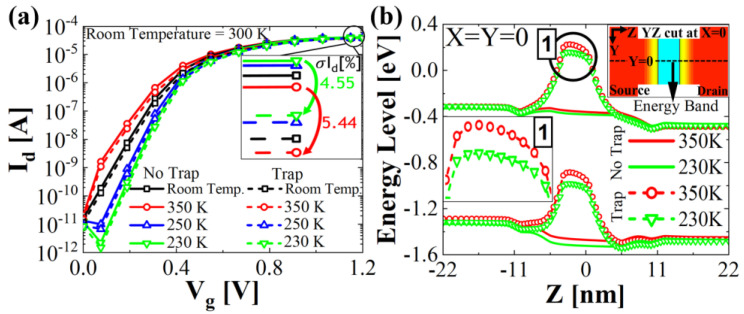
(**a**) *I_d_*−*V_g_* characteristics with different temperatures. (**b**) In the YZ plane cut, the energy band diagram at 230 K and 350 K with and without defect in the *X* = *Y* = 0 μm. The displacement defect is located at the worst position, as shown in Figure 3.

**Figure 5 micromachines-13-01276-f005:**
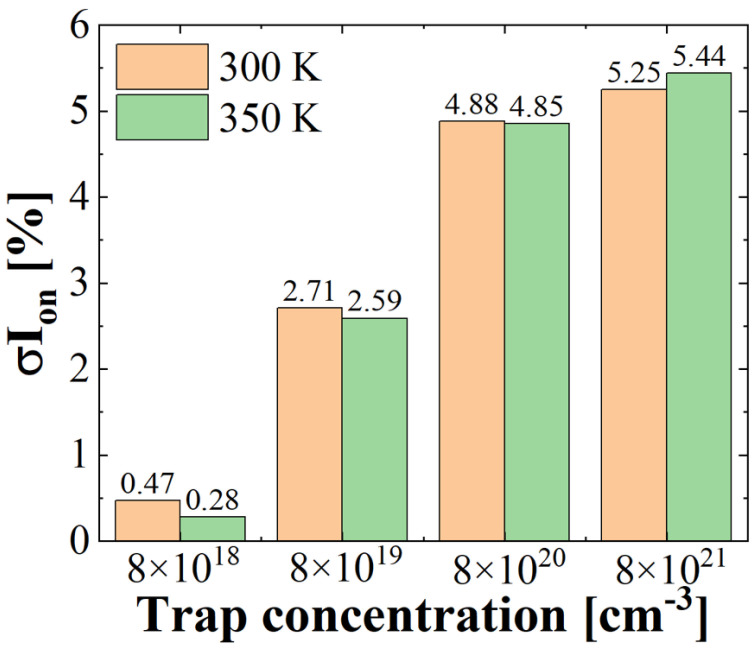
Rate of *I_on_* degradation with different temperatures and trap concentrations. Displacement defect is located at the worst position as shown in Figure 3.

**Figure 6 micromachines-13-01276-f006:**
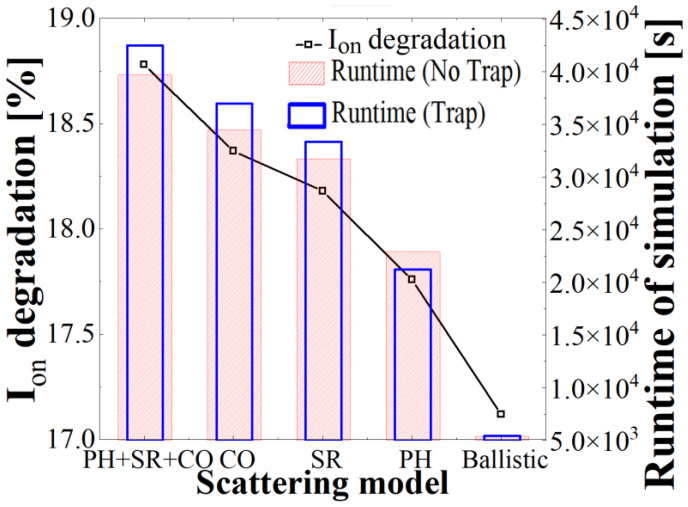
*I_on_* degradation owing to displacement defect and runtime of simulation with various scattering models. Various scattering models of coulomb (CO), surface roughness (SR), and phonon (PH) are compared regarding *I_on_* degradation and simulation runtime.

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
