# Peer review of "On-State Current Degradation Owing to Displacement Defect by Terrestrial Cosmic Rays in Nanosheet FET"

_micromachines, 2022, doi:10.3390/mi13081276_

Round 1
Reviewer 1 Report
Ha et al. presented a study on the non-ideal silicon-based Nanosheet FET (NS-FET) by using TCAD tool, with the inclusion of quantum transport effects. The relationship between the geometrical conditions, temperatures, trap concentrations, and scattering models were also considered as the variables for on-current reduction. The research is relevant to the nanoelectronics as the scaling of transistor is approaching the fundamental limits. I believe the work is publishable. However, there are a few comments to be addressed.
Below are my comments:
1. Line 84: This sentence 'with calibration of experimental reference [1]' is not clear to me. Is there any calibration done to the original data? If possible, could you also please provide the fitting graph for low Vd (0.05V)? The data low Vd is available in reference [1]. Is there any reason that the authors are focusing on high Vd (0.7V)?
2. As far as I know, the off-current is also important for the operation of a FET. Is the defect affecting only the on-current? Please provide justifications, if any.
3. Could you please include the equation on how the Ion degradation is computed?
4. Just a suggestion, you may also include a data set for temperature below 300K, to have more comparisons.
5. In Figure 5, the authors show data for simulation runtime, but I couldn't find any discussion on the simulation runtime. Is it something related to computational cost?
6. Line 162-163: "For the temperature effect, the pattern of Ion degradation in high and low temperatures is different with concentrations of displacement defect." Could you please explain what do you mean by different? Perhaps the sentence should be more specific and precise.
Author Response
Thank you for your valuable comment.
Please, find out the attachment for response to review comment.

Reviewer 2 Report
1. The novelty of this study is not clear. Authors needs to state the novelty clearly and concise.
2. Needs to add more background to explain how does defects will effects on FET performance.
3. In fig 1 (b) what is the value of drain-source voltage used to measure the I_d vs V_g?
4. in fig 2, there is no drain-source voltage reported. Its also important parameter to effect on device performance.
5. there is no comparison for the I_on degradation for the changing NS width and thickness with previous reported results. How does I_on degradation happens if NS thickness increase or decrease.
6. How does NS defects will effect on FET mobility and subthreshold swing?
7. Authors needs to describe the conductance of the NS FET and also current on/off ratio by changing the width and thickness with defects.
8. Conclusion needs to be improved.
Author Response

(The authors gave the same response as above.)

Reviewer 3 Report
Reference: micromachines-1834954
Review Report
The MS entitled “On Current Degradation Owing to Displacement Defect by Terrestrial Cosmic Rays in Nanosheet FET” by J. Ha et al. reports on a TCAD study of the effect of the defect displacement in NS-FETs (nanosheet field-effet transistors). Different parameters such as device geometry, temperature, and trap’s concentration were considered in the study.
The paper English level is acceptable, but authors must make sure an experienced translator will review the MS carefully before resubmission.
The problem addressed in the paper is very relevant and the TCAD software used in the MS is suitable to study the problem. In my opinion, the paper could be accepted if authors successfully address/answer the issues questions below:
1 – In Section 2. Materials and methods authors state:
“To consider the subband Boltzmann transport equation, Schrödinger equation, and Poisson equation (3D) coupled with the drift-diffusion transport model, a Sentaurus Device QTX instrument (quantum transport model of carrier quasi ballistic transport) was utilized”
Are authors considering that hot-electron effects (i.e. electrons traveling a substantial part of the channel while not being in equilibrium with the lattice) are negligible? I would expect some of the carriers being ballistic but not all them, accordingly, a model (such as the energy balance) taking into account carrier energy relaxation should be used instead of the drift-diffusion one that is only able to deal with momentum relaxation.
May authors comment on that issue and clarify it?
2 – Could authors comment on the values selected for the trap concentration?
3 – Do authors consider that studying low-temperature is not of interest taking into account the possible space applications?
4 – Please, could authors discuss in detail results on Figure 5? How was the problem modelled in each method (please, include explicitly the equations used in each model and give the values of the parameters used in the performed simulations). Also, please, discuss the degradation given in Figure 5. Why the Ion degradation seems to be only marginally dependent on the models (even using three of them simultaneously the degradation is very similar, close to 18.7%, to the one obtained using only the PH model (around 17.8%)? Additionally, even when assuming a ballistic model the degradation is not very different in value (17%), would that mean that mobility change is not the dominant mechanism in Ion degradation but charge trapping?
5 – Would it be possible to compare with experimental results by other authors?
6 – Please, give and analyze the variation of the threshold voltage with degradation.
7 – Finally, could authors obtain and plot transconductance and efficiency of the transconductance versus VGS (gate-to-source voltage)? This will clearly show the degradation.
Author Response

(The authors gave the same response as above.)

Round 2
Reviewer 1 Report
Thanks for addressing my comments.
Reviewer 3 Report
The question in my previous report were answered satisfactorily.